# A Zebrafish Mutant in the Extracellular Matrix Protein Gene *efemp1* as a Model for Spinal Osteoarthritis

**DOI:** 10.3390/ani14010074

**Published:** 2023-12-24

**Authors:** Ratish Raman, Mohamed Ali Bahri, Christian Degueldre, Caroline Caetano da Silva, Christelle Sanchez, Agnes Ostertag, Corinne Collet, Martine Cohen-Solal, Alain Plenevaux, Yves Henrotin, Marc Muller

**Affiliations:** 1Laboratory for Organogenesis and Regeneration (LOR), GIGA Institute, University of Liège, 4000 Liège, Belgium; ratish.raman@uliege.be; 2GIGA CRC In Vivo Imaging, University of Liege, Sart Tilman, 4000 Liège, Belgium; m.bahri@uliege.be (M.A.B.); christian.degueldre@uliege.be (C.D.); alain.plenevaux@uliege.be (A.P.); 3Hospital Lariboisière, Reference Centre for Rare Bone Diseases, INSERM U1132, Université de Paris-Cité, F-75010 Paris, France; cakaetano@hotmail.com (C.C.d.S.); agnes.ostertag@inserm.fr (A.O.); corinne.collet@aphp.fr (C.C.); martine.cohen-solal@inserm.fr (M.C.-S.); 4MusculoSKeletal Innovative Research Lab, Center for Interdisciplinary Research on Medicines, University of Liège, 4000 Liège, Belgium; christelle.sanchez@uliege.be (C.S.); yhenrotin@uliege.be (Y.H.); 5UF de Génétique Moléculaire, Hôpital Robert Debré, APHP, F-75019 Paris, France

**Keywords:** zebrafish, skeletal development, ECM, *efemp1*, notochord, vertebra, osteoarthritis

## Abstract

**Simple Summary:**

Osteoarthritis is a debilitating and painful joint disease affecting mainly aging animals and people. Previous results indicated that Efemp1, a protein present in the extracellular matrix that surrounds each cell, is increased in the blood, urine, and bone of osteoarthritic patients. We used the zebrafish as a model system to investigate the role of the Efemp1 protein in skeletal development and homeostasis. We showed that the *efemp1* gene is expressed in the brain, the pharyngeal cartilage, and the developing notochord which will later form the vertebral column. We generated a mutant in this gene, devoid of a functional Efemp1 protein, to show that this mutant presents transient deformities in its head cartilage at early stages. More importantly, adult mutants expressed a phenotype characterized by a smaller distance between vertebrae and ruffled edges (bone spurs) at the vertebral ends. This defect is reminiscent of that observed in spinal osteoarthritis; we therefore propose the *efemp1*−/− mutant line as the first zebrafish model to study this condition.

**Abstract:**

Osteoarthritis is a degenerative articular disease affecting mainly aging animals and people. The extracellular matrix protein Efemp1 was previously shown to have higher turn-over and increased secretion in the blood serum, urine, and subchondral bone of knee joints in osteoarthritic patients. Here, we use the zebrafish as a model system to investigate the function of Efemp1 in vertebrate skeletal development and homeostasis. Using in situ hybridization, we show that the *efemp1* gene is expressed in the brain, the pharyngeal arches, and in the chordoblasts surrounding the notochord at 48 hours post-fertilization. We generated an *efemp1* mutant line, using the CRISPR/Cas9 method, that produces a severely truncated Efemp1 protein. These mutant larvae presented a medially narrower chondrocranium at 5 days, which normalized later at day 10. At age 1.5 years, µCT analysis revealed an increased tissue mineral density and thickness of the vertebral bodies, as well as a decreased distance between individual vertebrae and ruffled borders of the vertebral centra. This novel defect, which has, to our knowledge, never been described before, suggests that the *efemp1* mutant represents the first zebrafish model for spinal osteoarthritis.

## 1. Introduction

Vertebrate skeletal development depends on transcription factors and signaling pathways controlling the differentiation and maturation of crucial cell types such as chondrocytes and osteoblasts [1]. These cells secrete the specific cartilage and bone extracellular matrix (ECM), respectively [2,3,4]. It is this ECM, made up of an organic and an inorganic component, that confers the mechanical and structural functions to the skeleton. The major collagens are obviously crucial for the structural integrity of the skeleton, as illustrated by mutations in their genes [5]. Other collagenous and non-collagenous proteins play additional roles in structuring and fine-tuning the functions of the skeletal ECM. Furthermore, increasing evidence shows that ECM proteins also play a role in controlling and shaping skeletal development [3,4]. ECM proteins such as osteocalcin [6], osteopontin [7,8], osteonectin (Sparc), and bone sialoprotein [9] shape the skeleton by binding calcium during mineralization but also by interacting with BMP, Wnt, or integrin signaling pathways.

Fibulins are highly conserved glycoproteins that can associate with numerous components of the extracellular matrix, such as the basement membrane and elastic microfibers [10]. Two subgroups of fibulins can be distinguished by the length and structure of their modules. The first subgroup is made up of lengthy fibulins (Fibulin-1, -2, -6 and -8) that have a propensity to form dimers. The second subgroup is comprised of short fibulins (Fibulin-3, -4, -5, and -7), all of which are monomeric forms. All fibulins have the same fundamental structure, composed of three domains, where the N-terminal domain I varies most amongst different members of the fibulin family. Domain-II, located centrally, is characterized by a varying number of EGF-like modules that contain calcium-binding sequences (cbEGF). Finally, the unique C-terminal domain-III consists of 120–140 amino acids and is also known as the fibulin-type module [11]. The short fibulins have an additional cbEGF-like module in domain-I, while the long fibulins contain three anaphylatoxin modules [12]. Though fibulins are close in terms of their structure and, to some extent, location, they have varied functions and binding partners [11]. They play an important part in tissue remodeling during embryonic development, in maintaining the structural integrity of basement membranes and elastic fibers, and in other cellular activities [13,14,15]. Some fibulins have been linked to tissue organogenesis, vasculogenesis, fibrogenesis, and cancer because of their participation in the production and stabilization of the ECM [16].

Fibulin-3, now renamed as EGF-containing fibulin extracellular matrix protein 1 (EFEMP1) [17], is highly expressed all over the body. It is most prevalent in tissues that are rich in elastic fibers and in ocular structures [18]. In humans, mutations in the *EFEMP1* gene cause Malattia Leventinese/Doyne honeycomb retinal dystrophy (ML/DHRD), a form of early onset macular degeneration [19]. They have also been linked in genome-wide association studies to a variety of complex phenotypes, including developmental anthropometric factors and defects in connective tissue function [20] or inguinal hernia [21]. In addition, alterations in EFEMP1 expression have been linked in humans to a variety of cancers [20]. In mice, EFEMP1 is expressed in the heart, lungs, placenta, skeletal muscle [14], and in the condensing mesenchyme that gives rise to bone and cartilage, suggesting its role in skeleton development [17]. An *Efemp1−/−* KO mouse displayed reduced fertility, premature aging, decreased body mass, lordokyphosis, as well as generalized fat, muscle and organ atrophy [22]. Hernias, including inguinal hernias were also observed, possibly resulting from a marked reduction of elastic fibers that was observed in the fascia, a thin connective tissue surrounding and protecting structures throughout the body. Interestingly, no macular degeneration was observed in these mice, while expression of a mutated version (R345W) of EFEMP1 did cause deposits in the retinal pigment epithelium [23,24]. Overexpression of EFEMP1 in mouse inhibits angiogenesis and chondrocyte differentiation by affecting the creation of cartilage nodes, as well as the production of proteoglycans [25]. EFEMP1 is also known to interact with the matrix-bound matrix metalloproteinases (MMPs) inhibitor, the basement membrane protein known as extracellular matrix protein 1 (ECM1) and tissue inhibitor of metalloproteinase-3 (TIMP-3) [26].

Osteoarthritis (OA) is a condition of increasing interest in aging populations that is characterized by joint pain, loss of articular cartilage, and sclerosis of the subchondral bone [27]. Recently, higher levels of EFEMP1 fragments (Fib3-1, Fib3-2, and Fib3-3) have been detected in the serum and urine of OA patients, thus representing potential biomarkers for screening OA [28,29,30]. In addition, secretome data revealed that sclerotic osteoblasts collected from OA subchondral bone secrete significantly higher amounts of the 3 EFEMP1 fragments than healthy tissue [31]. Furthermore, it was shown that EFEMP1 was highly expressed in sections of articular cartilage in knee joints from OA patients [32].

Here, we decided to investigate in more detail the function of EFEMP1 in vertebrate skeletal development, using the zebrafish (*Danio rerio*) as a model system. The zebrafish has recently become an excellent model system for studying teleost and mammalian skeletal development and homeostasis. Indeed, the major signaling pathways are conserved among vertebrates and many genes were shown to play similar roles in this species [33,34]. Here, we investigate for the first time the early expression pattern of the *efemp1* gene in zebrafish embryos, and we characterize the effect on skeletal development of a mutation in this gene, both in larval stages and adults.

## 2. Materials and Methods

### 2.1. Fish and Embryo Maintenance

Zebrafish (*Danio rerio*) were reared in a recirculating system from Techniplast (Buguggiate, Italy) at a maximal density of 7 fish/L. The water characteristics were as follows: pH = 7.4, conductivity = 50 mS/m, and temperature = 28 °C. The light cycle was controlled (14 h light, 10 h dark). Fish were fed twice daily with dry powder (ZM fish food^®^, Zebrafish Management Ltd., Winchester, UK) with size adapted to their age, and once daily with fresh nauplii from *Artemia salina* (ZM fish food^®^). Larvae aged less than 14 days were also fed twice daily with a live paramecia culture. Wild type from the AB strain and mutant lines were used. The day before breeding, two males and two females were placed in breeding tanks out of the recirculating system, with an internal divider to prevent unwanted mating. On the day of breeding, fish were placed in fresh aquarium water and the divider was removed to allow mating. Eggs were raised in E3 (5 mM Na Cl, 0.17 mM KCl, 0.33 mM CaCl_2_, 0.33 mM MgSO_4_, 0.00001% methylene blue).

### 2.2. In Situ Hybridization

In situ labelling was performed as previously described [35]. The probe was made for *efemp1* using nested PCR (first PCR: primers F: 5′-AGTACGGGTGTGTGAACAGC-3′: R: 5′-CACACTGCCTACTAGTGTTTCAGG-3′; nested: primer R: 5′-GCGAATTGTAATACGACTCACTATAGGGGCAACAGACAGAACGCAGAAG-3′) (655 bp covering part of the 3′-untranslated region) and the antisense probe RNA was synthesized via in vitro transcription using the DIG SP6/T7 Transcription kit Roche (Merck, Overijse, Belgium). In situ hybridization was performed as previously described [36]; the larvae were photographed under a stereomicroscope (Leica, Wetzlar, Germany) or a dissecting microscope (Olympus, Antwerp, Belgium)

### 2.3. Generation of Mutant Lines

The mutant line for *efemp1* was created using CRISPR/Cas9 as previously described [37,38]. The target site for the CRISPR guide RNA was AAGTGTATAAACCACTACGG, located in coding exon 3 of the *efemp1* gene. The generated deletion (line *efemp1^ulg074^*) induces a frameshift in the coding sequence at amino acid 62 and a STOP codon at position 77.

For genotyping, DNA was isolated from fin clips from adults or larvae at various stages of development in 50 mM NaOH via heating in a 95 °C water bath for 20 min. The mix was cooled down on ice for 10 min and the DNA extraction was stopped by adding Tris-HCl pH = 8.0, 1/5th the volume of NaOH, and spun down using a desktop centrifuge for 2 min. The extracted DNA was stored at 4 °C, or directly used for PCR. Primers for genotyping were F: 5′-CGAGTGTGTCCTCGTGTCTG-3′; R: 5′-CGTGGCAGTAGTTGTGTTGG-3′.

### 2.4. RNA Extraction and Sequencing

The RNA was extracted from dissected adult caudal complex using the RNA mini extraction kit (Qiagen, Hilden, Germany) according to the manufacturer’s instructions. The RNA was treated with DNAseI (Qiagen, Hilden, Germany) to avoid DNA contamination. The quantity and quality of each extract was assessed via nanodrop spectrophotometer measurements, then the RNAs were stored at −80 °C. The integrity of total RNA extracts was assessed using the BioAnalyzer (Agilent, Santa Clara, CA, USA). RIN (RNA integrity number) scores were >9 for each sample.

The cDNA libraries were generated from 100 to 500 ng of extracted total RNA using the “Stranded Total RNA Prep” kit (Illumina, San Diego, CA, USA) according to the manufacturer’s instructions. All cDNA libraries were sequenced on a NextSequ550 sequencing system (Illumina, San Diego, CA, USA), in 2 × 76 bp (paired end). Approximatively 20–25 M reads were sequenced per sample. The sequencing reads were processed through the Nf-core rnaseq pipeline 3.0 [39] with default parameters and using the zebrafish reference genome (GRCz11) and the annotation set from Ensembl release 103 (www.ensembl.org; accessed on 1 May 2020). Differential gene expression analysis was performed using the DESeq2 pipeline [40].

### 2.5. Alcian Blue (AB) Staining

AB staining is one of the most widely applied techniques for staining ECM glycosaminoglycans to observe cartilage structures. Larvae were sacrificed by exposure to MS-222 (Ethyl 3-aminobenzoate methane sulfonate; Merck, Overijse, Belgium) (0.048% *w*/*v*) at 5 dpf or 10 dpf. The larvae were fixed in para-formaldehyde (PFA) 4% for 14–16 h at 4 °C and thereafter rinsed three times with Phosphate Buffered Saline/Triton 0.1% for 10 min. The larvae were stained in 1 mL of alcian blue 8Gx (Sigma Aldrich, Hoeilaart, Belgium) at 0.04% alcian blue/10 mM MgCl_2_/80% EtOH pH 7.5 O/N, on low agitation. Thorough rinsing was performed at least 7 to 8 times with 80% EtOH/10 mM MgCl_2_, on low agitation till the excess of blue stain was washed and the washing solution appeared clear. The larvae were washed with 50% EtOH pH 7.5 for 5 min and then with 25% EtOH pH 7.5 for 5 min. Bleaching was performed by adding 6 mL of H_2_O_2_ 3%/KOH 0.5% for 30 min for 5dpf and 45 min for 10 dpf larvae, respectively. Then, washing was performed twice for 20 min with 1 mL 25% glycerol/0.1% KOH to remove the bleaching solution. Rinsing and destaining was performed thrice at 50% glycerol/0.1% KOH for 30 min. The solution was replaced with a fresh solution of 50% glycerol/0.1% KOH and stored at 4 °C. The larvae were placed in lateral or ventral view onto glycerol (100%) for imaging. Images of stained larvae (*n* = 20–30 larvae) in three or more independent experiments were obtained on a dissecting microscope (Olympus, cell B software, version 3.4).

### 2.6. Alizarin Red (AR) Staining

Larvae were sacrificed at 5 dpf and fixed in PFA 4% for 14–16 h at 4 °C and thereafter rinsed three times with Phosphate Buffered Saline/0.1% Tween (PBST) for 10 min. Bleaching was performed by adding 6 mL of H_2_O_2_ 3%/KOH 0.5% for 30 min for 5 dpf, followed by washing twice for 20 min with 1 mL 25% glycerol/0.1% KOH to remove the bleaching solution. The larvae were stained with AR (Merck, Overijse, Belgium) at 0.05% in the dark for 30 min on low agitation. Rinsing and de-staining was performed thrice at 50% glycerol/0.1% KOH for 30 min. The solution was replaced with a fresh solution of 50% glycerol/0.1% KOH and stored at 4 °C. The larvae were placed in lateral or ventral view onto glycerol (100%) for imaging. Images of stained larvae in two independent experiments were obtained on a binocular (Olympus, cell B software).

### 2.7. Image Analysis of Larvae Stained for Cartilage or Bone

Image analysis was performed on the pictures of larvae stained with alcian blue for cartilage or alizarin red for bone. According to Aceto et al., 2015 [41], cartilage (alcian blue) images were analyzed by measuring the distances from anterior to ethmoid plate, anterior to posterior (head length-hl), articulation left to articulation right (d-art), ceratohyal ext. left to ceratohyal ext. right (d-cer), ethmoid plate to posterior, hyosymplectic left to hyosymplectic right (d-hyo), and the angle formed by the two ceratohyals (a-cer). Bone (alizarin red) images were evaluated by estimating the degree of mineralization (absent, low, normal/intermediate, high) of the following elements [42]: maxillary (m), dentary (d), parasphenoid (p), entopterygoid (en), branchiostegal ray-1 (br1), opercle (o), ceratohyal (ch), and hyomandibular (hm) (see also Appendix A for illustration).

### 2.8. Micro-Computed Tomography Scanning (µCT)

WT and *efemp1*−/− zebrafish siblings were grown in the same tank at identical density to minimize variability before being analyzed with µCT. For quantitative evaluation, 6 wt and 6 *efemp1*−/− at 1 year old were selected to document the standard length and thereafter analyzed. The zebrafish were sacrificed and fixed overnight at 4 °C in 4% (*w*/*v*) PFA. Individual zebrafish were kept hydrated in a sponge covering and placed in a sample holder during µCT acquisitions (SKYSCAN 1272 scanner, Bruker Corporation, Kontick, Belgium).

Whole body scans were acquired at 70 kV and 100 µA with a 0.50 mm aluminum filter and at an isotropic voxel size of 21 µm. For high-resolution scans and quantitative analysis of the first precaudal vertebrae, zebrafish were scanned at 70 kV and 100 µA with a 0.5 mm aluminum filter at an isotropic voxel size of 7 µm. For all samples, ring artifact and beam hardening correction was kept constant, and no smoothing was applied during reconstruction (NRecon, Bruker). Reconstruction of the scans was performed using the NRecon version 2.0 software (Bruker Corporation) and resulted in a single dicom file for each voxel size 21 and 7 µm. Images with 7 µm voxel size were manually segmented using PMOD version 4.0 (PMOD Technologies, Zurich, Switzerland) to extract precaudal vertebrae and both vertebral thickness and vertebral length. GraphPad Prism9 was used to perform ordinary one-way ANOVA test for comparing wt controls versus mutants.

Further analysis of the 21 μm images was performed using the FishCuT version 1.2 Software [43,44]. Briefly, FishCuT is a matlab toolbox designed to analyze microCT images of zebrafish and extract morphological and densitometric quantitative information of zebrafish [43]. Since FishCuT was initially developed on images obtained with a vivaCT40 (Scanco Medical, Brüttisellen, Switzerland), we first adapted the parameters (intercept and slope) that should be used in the TMD conversion formula (https://doi.org/10.7554/eLife.26014, accessed on 7 February 2022). These parameters were estimated from the calibration scan performed on the same day of the data acquisition. FishCuT output data were then subjected to statistical analysis (multiple linear regression analysis with post hoc d’Agostino–Pearson normality testing) in GraphPad Prism9 software (9.4.1).

The following combinatorial measures were considered and quantified for each vertebra: centrum surface area (Cent.SA), centrum thickness (Cent.Th), centrum tissue mineral density (Cent.TMD), centrum length (Cent.Le), haemal arch surface area (Haem.SA), haemal arch thickness (Haem.Th), haemal arch tissue mineral density (Haem.TMD), neural arch surface area (Neur.SA), neural arch thickness (Neur.Th), neural arch tissue mineral density (Neur.TMD), vertebral surface area (Vert.SA), vertebral thickness (Vert.Th), and vertebral tissue mineral density (Vert.TMD). Vertebral measures (Vert) represent the total vertebral body, with all three elements (centrum, haemal arch, and neural arch) combined. Multivariate analysis was performed for statistical significance.

## 3. Results

### 3.1. efemp1 Expression in Zebrafish

To gain insight into the expression domain of the *efemp1* gene during early zebrafish development, we performed whole mount in situ hybridization experiments on 48 hours post-fertilization (hpf) zebrafish embryos (Figure 1). *efemp1* was expressed in the brain (br), in the pharyngeal region (pa), and in the notochord (nt) along the entire length of the trunk (Figure 1A,B). Closer inspection of the expression in the notochord revealed that it was seen in the chordoblasts (cb) (Figure 1C,D), responsible for secretion of the notochordal sheet (nts) [45,46]. No labelling was observed in 24, 96, or 120 hpf larvae.

### 3.2. Characterization of Early Skeletal Development in a Mutant in the efemp1 Gene

To gain insight into the function of the *efemp1* gene during development, we generated a mutant (*efemp1^ulg074^*) carrying a 5-nucleotide deletion (delin −7 + 2) at position 184 relative to the ATG, leading to disruption of the coding sequence at amino acid 62 and a STOP codon at position 77. Heterozygous parents carrying this mutation were crossed and their offspring larvae were stained for cartilage with alcian blue at 5 and 10 days post-fertilization (dpf).

Each larva was photographed, and its DNA was subsequently extracted for individual genotyping. Morphometric measurements were performed [41] and assigned to, respectively, wt and homozygous *efemp1*−/− mutants. Measurements on 5 dpf larvae revealed a significant (*p* = 0.032 and 0.047, respectively) decrease in the distance between the Meckel’s-palatoquadrate articulations (d-ar) and between the posterior end of the ceratohyals (d-cer), while the angle between ceratohyals (a-cer) (*p* = 0.19) and the head length (hl) (*p* = 0.76) were not affected. Thus, it appears that the chondrocranium was narrower in the *efemp1*−/− mutants at 5dpf (Figure 2A,C). The head width seemed to be restored with age, as in 10dpf larvae the d-ar and d-cer were not significantly different anymore (Figure 2B,D).

We also performed staining of the mineralized bones with alizarin red on 5 dpf larvae; no significant difference was observed on bone mineralization between wt and mutant larvae (Appendix A).

### 3.3. The Skeleton in Adult efemp1−/− Mutants

No impact was observed on the survival or growth of the *efemp1*−*/*− mutants relative to their wt siblings. Therefore, we grew them to 1.5 years in order to analyze their adult skeleton with µCT analysis [47]. No difference was apparent in the projected images of the µCT scans (Figure 3A). We then selected the precaudal vertebrae 6–8 (Figure 3B) for further morphometric analysis. In particular, we measured the vertebral length and the vertebral thickness (Figure 3C). No difference was observed in the vertebral length; however, the vertebral thickness was increased in the *efemp1*−*/*− mutants relative to wt in all three vertebrae, although never reaching significance (*p* < 0.05, *n* = 6) (Figure 3D). We further analyzed the vertebral column over its entire length by quantifying combinatorial measures for each vertebra. This analysis revealed that the tissue mineral density (TMD) was significantly increased (*p* = 0.04) in all vertebrae and in vertebral centra (*p* = 0.04) of *efemp1*−*/*− mutants compared to wt siblings, while all other bone properties were unaffected (Figure 3E).

Upon closer inspection of the images of the precaudal vertebrae 6–8, we observed a decrease in the intervertebral distance between individual vertebrae (Figure 4A). This distance was significantly reduced for both the vb06-vb07 (0.0051) and vb07-vb08 (*p*= 0.021, respectively) (Figure 4B). In addition, we also observed that the anterior and posterior ends of each vertebral body, facing the neighboring one, appeared to be ruffled in the mutants, compared to the smooth surface observed in the wt (Figure 4A).

## 4. Discussion

The ECM is a complex network made up of a variety of multidomain proteins that interact with each other in a specific manner to produce a composite stable structure [48]. These structures contribute to the mechanical properties of tissues and play a crucial role in controlling the most fundamental characteristics of cells, such as proliferation, adhesion, migration, polarity, differentiation, and apoptosis [49,50,51]. In that context, it is very important to understand and study the role of ECM proteins in skeletal development and homeostasis.

Among the non-collagenous ECM proteins, EFEMP1/FIBULIN3 has been shown in humans and mice to be expressed in a wide range of tissues, including cartilage and bone [17], and to be involved in numerous connective tissue diseases. Very little is known about Efemp1 in zebrafish and especially its role in skeletal development. Our study provides a first characterization of the *efemp1* gene in zebrafish development. In situ hybridization revealed *efemp1* expression in the head, pharyngeal region and in the notochord in 48hpf zebrafish embryos (Figure 1). Using the CRISPR/Cas9 gene editing method, we generated a mutant carrying a deletion of five nucleotides in the *efemp1* coding region, introducing a premature STOP codon and thus coding for a protein devoid of its major functional domains. Although we were not able to confirm the absence of Efemp1 protein in the mutants for lack of available antibodies, we did thoroughly genotype each individual larva and adult to identify the bona fide homozygous mutants before analysis. In addition, we showed via RNA-Seq that in adult mutant fish, the *efemp1* RNA was slightly decreased (log(fold-change) = −0.44, *p*-value = 0.27), indicating some extent of degradation of the mutant RNA. The described phenotypes were always shared by all the mutants, and not present in wt individuals for *efemp1*, excluding the involvement of an inadvertent off-target gene. We therefore believe that this mutation is actually causing the phenotype.

Our first morphometric characterizations of the *efemp1*−/− line during early stages revealed some effects in the head cartilage, with significant decreases in the distance between the lower jaw articulations (d-art) and some decrease in the ceratohyal angle (a-cer) at 5 dpf (Figure 2). These findings indicate a narrowing of the medial chondrocranium at 5 dpf, which, however, disappears at 10dpf, possibly due to some compensatory mechanism in the developing larvae. Bone structures were also not affected at these early stages.

In contrast, the 1.5-year-old *efemp1*−/− zebrafish display an increased TMD of the vertebral centra and the entire vertebrae, along with a slightly increased thickness in vertebral centrae 6–8. The most striking effect was, however, the significantly decreased distance between the vertebral bodies in all mutants. This reduced intervertebral disk space was concomitant of the appearance of ruffled edges, or bone spurs, at the extreme ends of the vertebral bodies (Figure 4). This phenotype is reminiscent of the osteoarthritic OA-like phenotype of the spine as described in humans [52,53,54]. OA in synovial joints is characterized by articular cartilage degeneration, synovial inflammation, and changes in the periarticular and subchondral bone [55]. There are many different locations within the body where an individual could possibly develop OA, including the leg, the synovial knee, ankle, wrist, elbow, shoulder, or hip joint [56], but also the facet joints of the vertebra in the spine [53,54,57]. The degeneration of the cartilage surface causes the formation of vertebral osteophytes, or bone spurs, followed by inflammation of the facet joints, ultimately causing narrowing of the intervertebral disc space [52].

OA studies in the zebrafish have been previously proposed [58,59,60,61,62]; however, they focused on the study of the jaw joint (the articulation between the palatoquadrate and the Meckel’s), while spinal deformities resembling osteoarthritis have been previously described in aging zebrafish [63]. Taken together, these observations indicate that the *efemp1*−/− mutant described here represents the first zebrafish model for spinal osteoarthritis. Recently, it was shown that both age-related and experimentally induced osteoarthritis in the knee joints was more severe in *Efemp1−/−* mice [32], further supporting our proposal that the *efemp1*−/− zebrafish constitute a valid model for studying the pathogenesis and putative treatments of vertebral osteoarthritis.

Although we found that the early skeletal development was largely unaffected in the *efemp1*−/− mutant, our results indicate that the mutation affects the health of the vertebral column at later stages. In this context, the early expression of *efemp1* in the zebrafish notochord is interesting, as closer inspection revealed that the expression takes place in the chordoblasts, surrounding the notochord and responsible for secreting the notochordal sheath [64]. This structure is known to encase the notochord and comprises of three layers: a thin inner layer of elastin, a thick layer containing lamellar collagen type II, and an outside layer also made of elastin, the *elastica externa*. The chordoblasts are found at the level of the collagen type II fibers, while osteoblasts and collagen type I are located on the outside layer [45]. Later, in teleost species like zebrafish, the formation of vertebral centra (chordacentra) takes place in the absence of cartilage via the mineralization of the notochordal sheath [65]. This mineralization is initiated by chordoblasts, which are derived from the notochord and not from sclerotome-derived osteoblasts [66,67]. It is tempting to speculate that the *efemp1* expression in early chordoblasts would be able to affect the health of the vertebral column in adults. However, at this time, it is unclear how the *efemp1* mutation affects the structure of elastic fibers in the notochordal sheath, or precisely how it leads to an increased vertebral thickness, vertebral TMD, and the phenotype of spinal OA in older individuals. Further investigations into histological changes at various stages, as well as changes in signaling pathways, will be required to better understand the onset of spinal OA in this model.

## 5. Conclusions

Taken together, our results show that a mutation of the *efemp1* gene in zebrafish causes transient deformities in chondrocranium at 5dpf, which, however, disappear at later stages. Interestingly, µCT analysis of 1.5-year-old mutants revealed that the distance between individual vertebrae was reduced in the mutants, along with the presence of a ruffled border, indicative of bone spurs. This defect very much resembles that observed in human spinal osteoarthritis, making this mutant the first zebrafish model for this condition. Zebrafish may indeed be the better animal model for spinal osteoarthritis in fish and bipedal land animals, as the loading direction on the vertebral column is axial in fish [68], similar to humans and in contrast to other, quadruped rodent models. It is, at present, unclear how far the increased TMD that was also observed in the adult mutants plays a role in the onset of the spinal OA condition. Similarly, the role of *efemp1* expression in the early chordoblasts, possibly via strengthening the elastic properties of the vertebral sheath, will need to be investigated in future research.

## Figures and Tables

**Figure 1 animals-14-00074-f001:**
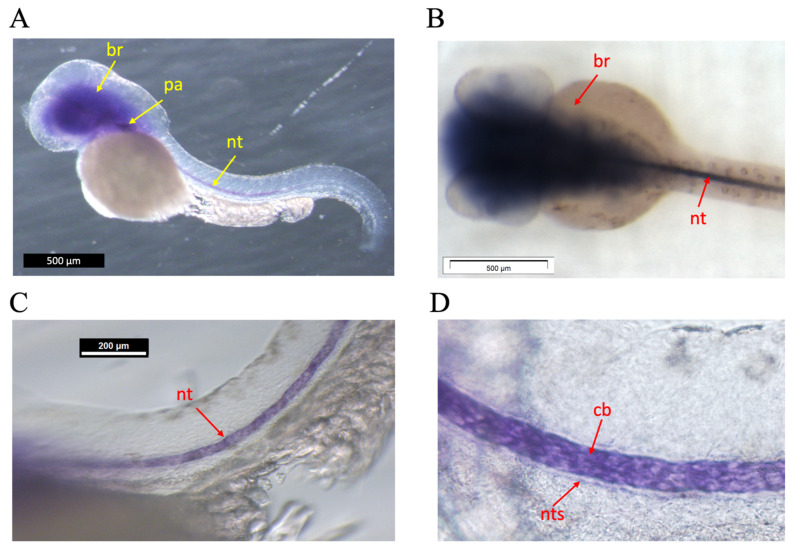
Whole mount in situ hybridization on 48 hpf zebrafish embryos. (**A**) Lateral view, anterior to the left. Expression is seen in the brain (br), the pharyngeal area (pa), and in the notochord (nt) (**B**) Dorsal view, anterior to the left. Dissection microscope picture. Expression is seen in brain and notochord. (**C**,**D**) Lateral views, anterior to the left. Enlarged view of expression in the notochord, specifically (**D**) in the chordoblasts (cb) immediately adjacent to the notochordal sheath (nts). Other, non-related probes served as negative control.

**Figure 2 animals-14-00074-f002:**
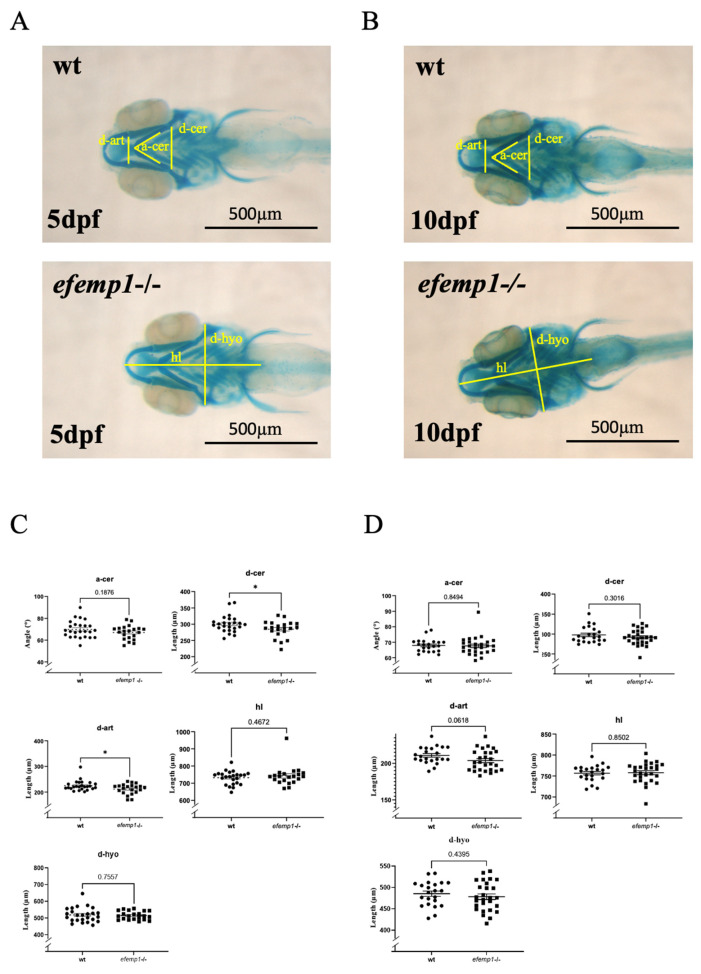
Cartilage staining with alcian blue of 5 dpf (**A**,**C**) and 10 dpf (**B**,**D**) wt and *efemp1*−/− mutant larvae. The different measures are illustrated in (**A**,**B**): d-ar: distance between the Meckel’s-palatoquadrate articulations; d-cer: distance between the posterior ends of the ceratohyals; a-cer: angle between the ceratohyals; hl: head length and d-hyo: head width. (**C**) *efemp1*−/− reveal a reduced distance between the articulation (d-ar) and narrower distance between the ceratohyal elements (d-cer) at 5 dpf compared to wt (wt *n* = 24, *efemp1−/− n* =21). (**D**) No difference in the cartilage elements in *efemp1*−/− at 10dpf compared to wt (wt *n* = 22, *efemp1−/− n* = 27). The graphs show the individual data points, the mean value ± SEM; significance: * *p* < 0.05 (unpaired student’s *t*-test).

**Figure 3 animals-14-00074-f003:**
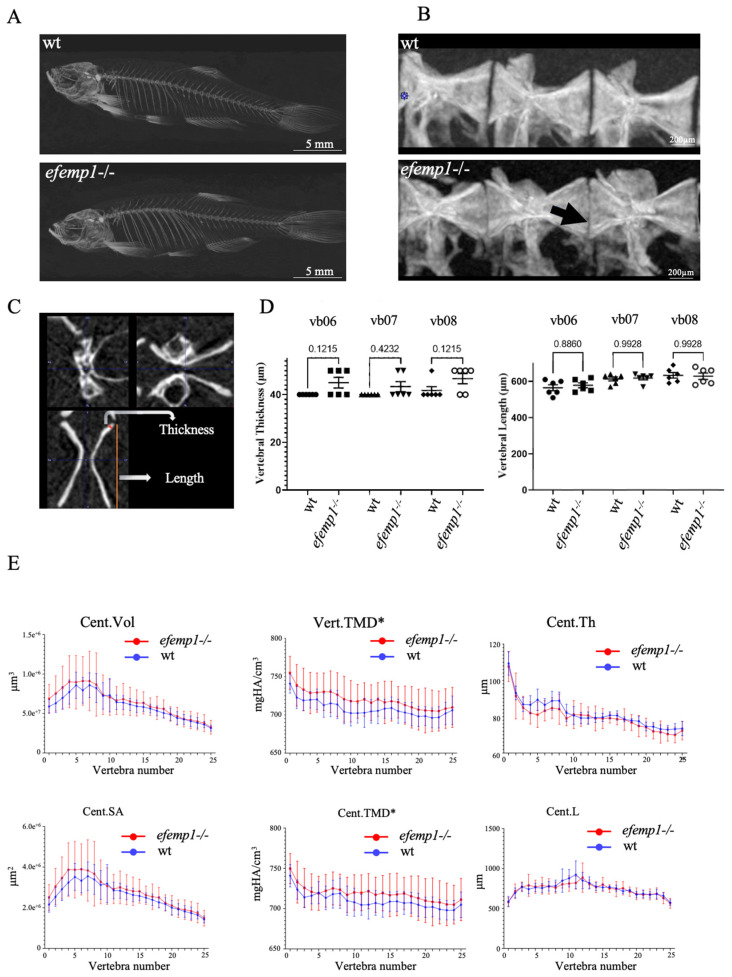
Increased TMD in *efemp1*−/− mutants. (**A**) Representative µCT scans (MIPi = Maximum Intensity Projected image) of a 1.5-year-old wt (top) and *efemp1*−/− (bottom) adult. (**B**) Lateral view of pre-caudal vertebrae 6–8 (L to R) for the two groups; wt and *efemp1*−/−. The black arrow points to the decreased intervertebral distance and ruffled border. (**C**) Representative µCT scan of a vertebra in three planar views, showing the two morphometric measurements: vertebral thickness (µm) and vertebral length (µm). (**D**) Morphometric analysis comparing vertebral thickness and vertebral length of individual precaudal vertebral body numbers 6–8 (*n* = 6 fish/group) in *efemp1*−/− compared to wt. The values are expressed as mean ± SEM (standard error on mean), statistical significance as determined with ordinary one-way ANOVA test. (**E**) Line plots generated using the GraphPad Prism9 Software (v.9.4.1) of the data points obtained from the FishCuT Software revealing significantly increased TMDs in the entire vertebrae (Vert.TMD) and in the vertebral centra (Cent.TMD) of *efemp1*−/− adults relative to wt (*p* < 0.0001), with no significant differences observed in other combinatorial measures, (*n* = 6 fish/group and total no. of vertebrae analyzed = 25/individual). The values are expressed as mean ± SEM, significance: * *p* < 0.05, as determined via multiple linear regression analysis.

**Figure 4 animals-14-00074-f004:**
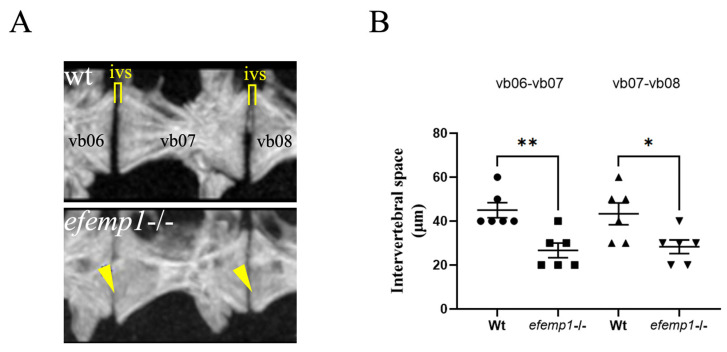
Reduced intervertebral disk space and bone spurs observed in the spine of *efemp1*−/− adult zebrafish. (**A**) Closeup view of pre-caudal vertebrae 6–8 (L to R) for wt and *efemp1*−/−, clearly showing reduced intervertebral disk space and bone spurs, indicated by yellow bar and arrowheads, respectively. (**B**) Intervertebral disk space calculated between vb06-vb07 and vb07-vb08 (*n* = 6 fish/group), revealing significant reduction of the intervertebral disk space in the *efemp1*−/− zebrafish adults. The values are expressed as mean ± SEM, significance: * *p* < 0.05, ** *p* < 0.01, as determined by ordinary one-way ANOVA test.

## Data Availability

All photographs used for the generation of this manuscript will be made publicly available (https://hdl.handle.net/2268/310280, accessed on 23 December 2023).

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
