# Peer review of "A Zebrafish Mutant in the Extracellular Matrix Protein Gene efemp1 as a Model for Spinal Osteoarthritis"

_animals, 2023, doi:10.3390/ani14010074_

Round 1

Reviewer 1 Report

Comments and Suggestions for Authors

In the submitted manuscript, the authors analyzed a Zebrafish KO of the efemp1 gene, mainly expressed in the extracellular matrix, suggesting its potential for osteoarthritis studies. The research concludes that µCT analysis of 1.5-year-old mutants revealed reduced vertebrae distance and a ruffled border, resembling human spinal osteoarthritis. This zebrafish model holds promise for studying the condition in diverse animals.

Overall, the manuscript is well-written. The conclusions are well-supported by the data and align with related work in the field, highlighting the findings' significance. The figures are clear and informative. However, I would advise the authors to discuss certain limitations of the study. There is a missing figure or data specifically demonstrating the mutant's loss of gene expression or protein in the extracellular matrix. I have no further suggestions for improvement.

Reviewer 2 Report

Comments and Suggestions for Authors

Comments and Suggestions for Authors

The study by Raman, et al., attempts to establish an efemp1 zebrafish mutant as a model for spinal osteoarthritis. Specifically, they used CRISPR/Cas9 to truncate Efemp1 to demonstrate its role in the health of the vertebral column at later stages. The authors associate this with the expression of efemp1 in notochord chordoblasts.

All together, there are some interesting observations in this manuscript. 

However, some rewriting is necessary. At this point the paper is somewhat difficult to read due to the number of run on sentences. For example (lines 42, 67, 77, 104-7, 291-3). There is also inconsistency with the placement of citations. The author should also examine all figure legends for completeness. For example, Figure 1 legend does not define (cb) chordoblasts. It is not clear which genes or species the author is referring to in (lines 104 – 7). There is no data to support the statement made in lines (246 – 8). This should be removed. The reference appears to be for the method. The writing is not precise and thereby confusing.

Second, some results are not convincing.

The authors use in situ hybridization in Figure 1 to show expression of efemp1 at 48 hpf in the brain, pharyngeal region and notochord.  The author states that there is no expression in the eye. This is not completely evident. The data would be more convincing regarding specific staining if section in situs of the various regions were provided. Further, the authors are trying to associate the expression of efemp1 in chordoblasts with abnormal vertebral column structure. Either section in situ, or immunofluorescence should be used to demonstrate efemp1 expression in chordoblasts. The data in Figure 3 (error bars) shows that there was a large amount a variation between mutant zebrafish. Is the phenotype due to the efemp1 mutation? The author should provide some explanation.

Comments on the Quality of English Language

 At this point the paper is somewhat difficult to read due to the number of run on sentences. For example (lines 42, 67, 77, 104-7, 291-3). There is also inconsistency with the placement of citations. The author should also examine all figure legends for completeness. For example, Figure 1 legend does not define (cb) chordoblasts. It is not clear which genes or species the author is referring to in (lines 104 – 7). There is no data to support the statement made in lines (246 – 8). This should be removed. The reference appears to be for the method. The writing is not precise and thereby confusing.

Reviewer 3 Report

Comments and Suggestions for Authors

Line 45: correct references

Line 70: correct references

Line 78: delete in humans

Line 79: delete bracket

Line 80: in which species?

Paragraph 2.2: add the amplicone size detect by the pcr primer. Add code number of T7 RNA polymerase

Line 147: add the dose of MS222

Line 149: write in full PFA

Line 150: write in full PSBT

Line 150: add manufacturer informations of Alcian blue

Line 163: add a paragraph for Alizarin red staining

Line 176: delete bracket after scanner

Figure 1: B better in the dorsal view images, position the animals correctly, if necessary remove the yolk

Add ISH af different time point: 24-72-96-120 hpf

Add sense probe as control for ISH for all time point

Line 246: move there fig 2

Line 248: add the novel data of alizarin

Line 261: move there fig 3

FIG 3: add all letters in the figure

Comments on the Quality of English Language

Minor editing of English language required

Round 2

Reviewer 2 Report

Comments and Suggestions for Authors

The study by Raman, et al., attempts to establish an efemp1 zebrafish mutant as a model for spinal osteoarthritis. Specifically, they used CRISPR/Cas9 to truncate Efemp1 to demonstrate its role in the health of the vertebral column at later stages. The authors associate this with the expression of efemp1 in notochord chordoblasts.

All together there are some interesting observations in this manuscript. The study provides data indicating efemp1 zebrafish mutants present transient deformities in its head cartilage at early stages. Adult mutants show smaller distances between vertebrae and ruffled edges (bone spurs) at the vertebral ends reminiscent of that observed in spinal osteoarthritis. The data are relevant for the field. The conclusions are consistent with the data and arguments presented.

Reviewer 3 Report

Comments and Suggestions for Authors

I would like to thanks authors for responding to my reviews